# Nanomaterials with Excellent Adsorption Characteristics for Sample Pretreatment: A Review

**DOI:** 10.3390/nano12111845

**Published:** 2022-05-27

**Authors:** Wen-Xin Liu, Shuang Song, Ming-Li Ye, Yan Zhu, Yong-Gang Zhao, Yin Lu

**Affiliations:** 1College of Environment, Zhejiang University of Technology, Hangzhou 310014, China; lwx187221@163.com (W.-X.L.); ss@zjut.edu.cn (S.S.); 2College of Biological and Environmental Engineering, Zhejiang Shuren University, Hangzhou 310015, China; yemingli@zjsru.edu.cn; 3Department of Chemistry, Zhejiang University, Hangzhou 310027, China; zhuyan@zju.edu.cn

**Keywords:** carbon nanomaterial, porous nanomaterials, magnetic nanomaterials, sample pretreatment, solid phase extraction

## Abstract

Sample pretreatment in analytical chemistry is critical, and the selection of materials for sample pretreatment is a key factor for high enrichment ability, good practicality, and satisfactory recoveries. In this review, the recent progress of the sample pretreatment methods based on various nanomaterials (i.e., carbon nanomaterials, porous nanomaterials, and magnetic nanomaterials) with excellent adsorption efficiency, selectivity, and reproducibility, as well as their applications, are presented. Due to the unique nanoscale physical–chemical properties, magnetic nanomaterials have been used for the extraction of target analytes by easy-to-handle magnetic separation under a magnetic field, which can avoid cumbersome centrifugation and filtration steps. This review also highlights the preparation process and reaction mechanism of nanomaterials used in the sample pretreatment methods, which have been applied for the extraction organophosphorus pesticides, fluoroquinolone antibiotics, phenoxy carboxylic acids, tetracycline antibiotics, hazardous metal ions, and rosmarinic acid. In addition, the remaining challenges and future directions for nanomaterials used as sorbents in the sample pretreatment are discussed.

## 1. Introduction

In analytical chemistry, the sample pretreatment methods are very important because they play a vital role in the whole analytical system. The complete sample analysis includes five steps: sample collection, sample pretreatment, instrumental analysis, data processing, and reporting results. The most time-consuming of these five steps is sample pretreatment, which accounts for about 60% of the whole process [1,2,3]. At present, solid phase extraction (SPE), with higher adsorption rate and better reproducibility, is widely used in the preconcentration of low concentration substances and the purification of complex samples [4]. As well as the SPE methods, various efficient and environmentally friendly pretreatment methods based on SPE, including solid phase microextraction (SPME), dispersed solid phase extraction (dSPE), magnetic solid-phase extraction (MSPE), the QuEChERS (quick, easy, cheap, effective, rugged, and safe) method [5,6,7,8], and the PRiME (process, robustness, improvements, matrix effects, and ease of use) pass-through cleanup procedure [9,10,11,12], have been developed. In these sample pretreatment methods, adsorbent with uniform particle size distribution, good dispersion, and high-efficiency adsorption capacity is the key factor for the performance of sample pretreatment.

With the continuous study of nanomaterials, researchers have found that nanomaterials have great potential in the field of sample pretreatment. The typical particle size range of nanomaterials is between 1~100 nm [13]; however, it does not mean that only the particle size between 1~100 nm is called nanomaterial, as long as there is a dimension within this range, then this kind of material belongs to nanomaterial. Nanomaterials are ideal solid phase adsorption materials [14,15]. Compared with general adsorbents, nanomaterials have the following characteristics: (1) smaller particle size and larger specific surface area, which increases the interaction between particles and analytes, thus improving the ability of nanomaterials to adsorb and separate analytes [16,17]; (2) very high surface energy and diffusivity, the nanoparticles can be in full contact with each other, the adsorption equilibrium can be achieved in a short time, and the nanomaterials have unique adsorption properties for metal ions and organic pollutants [18,19,20,21]; and (3) easy to be functionalized, and specific functional groups can be introduced to achieve efficient and selective extraction of target compounds. Due to so many excellent properties, nanomaterials have been used as adsorbents in various branches of analytical chemistry [22,23], such as drug analysis, biomedicine, environmental protection and forensic medicine. It is used for qualitative and quantitative analysis of various analytes in complex samples [24,25]. For example, polymer nanomaterials have been used to analyze benzene series, p-hydroxybenzoate, and imidazolinone in soil and sediment samples. The metal and mixed oxide nanomaterials [26,27,28], carbon-based nanomaterials [29,30,31,32], and magnetic nanomaterials [33,34,35] have been used to analyze substances in environmental water samples.

Nanomaterials can be divided into organic nanomaterials, inorganic nanomaterials, and organic–inorganic hybrid nanomaterials according to their composition [36]. In recent years, there is increasing research on hybrid composites, especially the magnetic hybrid composites. Magnetic nanoparticles have the characteristics of superparamagnetism, high adsorption capacity, a large specific surface area, and so on [37,38]. These characteristics make it more suitable to be adsorbent for SPE. Magnetic nanoparticles can be separated quickly by a magnetic field due to the superparamagnetism; therefore, when the analyte interacts with the magnetic nanoparticles, it can be easily separated from the complex samples by magnetic separation.

Recently, prominent progress has been obtained in nanomaterials research. In this article, we review the recent advances in the nanomaterials and discuss their classification and preparation, as well as strategies to improve their adsorption efficiency, selectivity, reproducibility, and applications in sample pretreatment. We will also pay considerable attention to the recent development in the sample pretreatment methods, with a focus on nanomaterials. Based on the applications of nanomaterials for the extraction of various chemicals (i.e., organophosphorus pesticides, phenoxy carboxylic acids, hazardous metal ions, etc.), we also provide some perspectives on the future trends of nanomaterials for the further design of novel sample pretreatment methods and beyond.

## 2. Synthesis of Nanomaterials and Their Applications in Sample Pretreatment 

### 2.1. Carbon Nanomaterials

Carbon nanomaterials have always been the focus of research, because carbon nanomaterials have good adsorption, electrochemical, catalytic, and gas storage properties. Due to the different bonding modes between carbon atoms, the structure and properties of carbon nanomaterials are different. Carbon nanomaterials can be divided into the following three basic types: zero-dimensional nanoparticle clusters, such as fullerenes (C_60_), successfully prepared by Kroto and co-workers in 1985 [39]; one-dimensional carbon nanotubes and carbon nanofibers, such as carbon nanotubes, which were ws discovered by Iijima in 1991 [40]; two-dimensional carbon nanolayers or membrane materials, such as graphene, which was discovered by Novoselov and co-workers [41]. With the continuous discovery of carbon nanostructures, and all of them showing very good adsorption properties and outstanding mechanical properties, researchers have a strong interest in them [42,43,44].

#### 2.1.1. Application of Fullerene in Sample Pretreatment

Fullerene is a hollow particle cluster composed of carbon atoms, which is very similar to graphene in structure. Graphene is composed of six-membered rings, whereas fullerene contains not only six-membered rings, but also five-membered rings, or even seven-membered rings [45]. Due to the different number of atoms that make up the particle group, fullerene has many isomers, but the most stable one is C_60_; therefore, C_60_ is the most common and widely studied and applied. As with a football, the molecular structure of C_60_ is also a spherical structure composed of thirt-two faces, of which twenty, six-membered rings, and twelve, five-membered rings, are connected by sixty carbon atoms; therefore, C_60_ is also called the football alkene [45,46]. 

In 2007, a stationary phase of C60 fullerene combined with silica for SPE was prepared by Rainer and co-workers [47]. The preparation method was probably an adsorption material prepared by linking silica and C_60_ through covalent bonds with aminopropyl as a linking agent. The C_60_-fullerene silica had a strong retention ability for small molecules and hydrophilic molecules; therefore, the new material has a high adsorption capacity for proteins and phosphopeptides. In addition, C60-fullerene silica was also used for the extraction of flavonoids, and the results showed that the recovery rate of flavonoids by C_60_-fullerene silica SPE was 99%.

In 2019, a new type of fullerene (C_60_), a soluble eggshell membrane protein (SESMP) composite, was prepared by Alheety and co-authors [48]. The Fe_3_O_4_-NPs was grafted onto SESMP to prepare a SPE packing for arsenic in six kinds of crude oil and environmental water samples (Figure 1). Under the optimized conditions, the limit of detections of As (III) and As (V) were 0.0473 ng/mL and 0.0325 ng/mL, respectively, and the relative standard deviation (RSD) of As (III) and As (V) are 1.15% and 1.31%. The results showed that the SESMP grafted with Fe_3_O_4_ NPs had a good extraction of arsenic in crude oil and environmental water samples. 

#### 2.1.2. Synthesis of Carbon Nanotubes and Its Application in Sample Pretreatment

A carbon nanotube is a kind of quasi-one-dimensional tubular macromolecule with a helical period and hollow cavity structure, and its surface mainly shows a hexagonal network structure, which is also mixed with pentagons or heptagons. The carbon nanotubes can be seen as coiled from a hexagonal network of graphite sheets [49]. According to the number of graphite layers, it can be divided into single-walled carbon nanotubes (SWCNTs) and multi-walled carbon nanotubes (MWCNTs). Carbon nanotubes have high thermal capacity, high chemical stability, a large specific surface area, small pore size, and easy modification. It has better adsorption properties compared with traditional adsorbents; therefore, carbon nanotubes are often used as adsorbents in sample pretreatment. However, carbon nanotubes cannot selectively adsorb the target, it is necessary to functionalize carbon nanotubes to improve their selectivity. There are three main methods for the preparation of carbon nanotubes: arc discharge, laser ablation, and chemical vapor deposition (CVD).

(1) The arc square method was born in 1991. During the arc synthesis of graphite by arc method under a high resolution transmission electron microscope, MWCNTs were found for the first time [40]. Arc discharge is the most traditional method for the preparation of carbon nanotubes. Its basic principle is that under certain conditions, a large number of electrons are produced under the field electron emission effect and hot electron emission effect between the cathode and anode, which bombard the anode. The temperature of the anode increases and causes the anode to evaporate. These evaporated carbon atoms are quickly reorganized in the process of cooling to form carbon nanotubes. Although the equipment used by arc discharge method is complex, and the process parameters are difficult to control, the carbon nanotubes prepared by this method have the advantages of few structural defects, good flexibility, straight tube, high crystallinity, and so on [50,51,52,53,54].

(2) In 1995, Smalley team [55] successfully demonstrated the preparation of carbon nanotubes by laser ablation for the first time. Its principle and mechanisms are similar to arc discharge. Their difference lies in the different methods of making carbon atoms evaporate. Arc discharge makes anode carbon atoms evaporate by releasing a large number of electrons that bombard the anode, whereas the law of laser ablation is to vaporize carbon atoms by bombarding graphite particles containing catalysts with high-energy laser beams, then reassembling them to form carbon nanotubes, and finally depositing them on the collector with the airflow [56]. 

(3) The above two methods are mainly used in the laboratory, and the large-scale production of carbon nanotubes is mainly through CVD. The principle of CVD is that hydrocarbons are pyrolyzed into hydrogen and carbon atoms under the action of transition metal catalysts at appropriate ambient temperature, and then these carbon atoms are reassembled to form carbon nanotubes; however, there are two situations in the growth process of carbon nanotubes: (i) there are metal particles at the tip of carbon nanotubes; (ii) there are metal particles at the roots of carbon nanotubes. The reason for the different grounding position of metal particles is due to the different intensity of metal–carrier interaction. When the metal–carrier interaction is strong, metal particles will appear at the root of carbon nanotubes; when the metal-carrier interaction is weak, metal particles will appear at the tip of carbon nanotubes. The overall performance of carbon nanotubes prepared by this method is not as good as that of carbon nanotubes prepared by the arc discharge method and laser ablation method, but a single yield of the arc discharge method and laser ablation method is lower than that of the CVD method; therefore, CVD is widely used in the industrial production of carbon nanotubes [57,58,59].

In 2017, Feist and co-workers [60] developed a simple and effective method for the determination of lead, cadmium, zinc, manganese, and iron in white rice and wild rice samples by using oxidized MWCNTs as adsorbents. The limit of detection was between 0.13 ng/mL and 0.35 ng/mL. The new preconcentration method was successfully applied to food analysis with an accuracy of less than 7%. This method can be applied to the preconcentration of Pb (II), Cd (II), Zn (II), Mn (II), and Fe (III) ions in rice samples. 

In 2018, Khamirchi et al. [61] reported novel MWCNTs modified by [2-(5-Bromo-2-pyridylazo)-5-(diethylamino)phenol] (Br-PADAP) for the preconcentration of a uranium ion. The result showed that the Br-PADAP-modified MWCNTs had much higher adsorption capacity for uranium ion than the original carbon nanotubes; therefore, the multi-walled MWCNT carbon nanotubes, modified by Br-PADAP, can be used as a SPE adsorbent for efficient adsorption of uranium in water treatment. Moreover, the enrichment factor of the prepared SPE column for the analysis of trace uranium in different environments was 300 times, and the interference of other ions to the pre-enrichment of uranium was minimal. The limit of detection was 0.14 μg/L and the RSD was about 3.3%.

#### 2.1.3. Synthesis of Graphene and Its Application in Pretreatment

In the 20th century, the theory of “two-dimensional crystal structure can not exist because of its thermodynamic instability” was not overturned until Geim and others successfully prepared graphene by mechanical exfoliation. Graphene is formed by the close accumulation of carbon atoms, showing a honeycomb lattice structure, which is a two-dimensional crystal structure [62]. Both fullerenes and carbon nanotubes can be made by wrapping or crimping graphene. At present, graphene has become one of the most widely used carbon materials. Due to its special structure, graphene has good chemical stability, good mechanical strength, and a high specific surface area. Moreover, graphene also has outstanding corrosion resistance and flexibility [63,64]; therefore, graphene is more suitable for the preparation of adsorbents than carbon nanotubes and fullerenes, especially for graphene oxide (GO), because there are oxygen-containing functional groups on GO, such as –OH, –COOH, C–O–C. Moreover, the application of GO to the preparation of composites will make the prepared composites have a better adsorption efficiency and faster adsorption kinetics [65,66]. Graphene shows a great application prospect in pre-enrichment, and has become one of the most widely used carbon nanomaterials in sample pretreatment. The preparation of graphene is mainly divided into physical methods and chemical methods. Mechanical stripping is the most traditional method in terms of physical methods, and CVD is the most widely used method in terms of chemical methods.

(1) The mechanical stripping method is to remove graphene from graphite by tape or ultrasonic dispersion. In 2004, graphene stripped from graphite using adhesive tape was done for the first time by Novoselov and co-workers [41]. The tape with graphene, and the silicon wafer coated with silicon dioxide film, were put into the solution to adsorb some thin graphene sheets on the silicon wafer by van der Waals force, so as to achieve the purpose of separating the graphene layer. The process of mechanical stripping is simple, and the graphene prepared by mechanical stripping has few structural defects, but its yield is low, and the uniformity cannot be accurately controlled.

(2) The principle of preparing graphene by CVD is that the carbon-containing compounds are decomposed into carbon atoms at a high temperature, and then these separated carbon atoms are deposited on the surface of the matrix to obtain graphene. In 2006, Somani and co-workers [67] separated carbon atoms by putting camphor in the ambient atmosphere of Ar, baking it at 180 °C, and finally depositing it on the surface of metal Ni to get graphene. CVD has the potential to be produced on a large scale. Moreover, the graphene prepared by this method can maintain good uniformity in the case of large area [68,69].

In 2021, Silvestro and co-workers [70] adjusted the hydrophilicity of chitosan-based membranes by introducing different amounts of GO in order to obtain multifunctional materials used as adsorbents in SPE. The results showed that the introduction of GO indeed reduced the hydrophilicity of the composite membrane and increased the adsorption capacity of hydrophobic pollutants. 

In 2021, a novel molybdenum disulfide–GO composite (MoS_2_/GO) was synthesized by Wenjing and co-workers [71], which was used as an adsorbent for dispersive SPE (Figure 2). MoS_2_/GO was used for the adsorption of four preservatives (i.e., methylparaben, ethylparaben, propylparaben, and butylparaben) prior to high performance liquid chromatography (HPLC) analysis. Under optimal conditions, the results showed that MoS_2_/GO has a strong enrichment ability for these preservatives. The RSD obtained by this method was less than 8.0%, and the limit of detection was between 0.4 ng/mL and 2.3 ng/mL. It can be seen that MoS_2_/GO has broad prospects in the determination of parabens preservatives.

### 2.2. Porous Nanomaterials

Porous materials can be divided into three categories according to the pore size: macroporous materials (pore size greater than 50 nm), mesoporous materials (pore size between 2 and 50 nm), and microporous materials (pore size less than 2 nm) [72,73,74,75]. It can also be classified according to its composition, which can be divided into inorganic porous materials, organic porous materials and inorganic–organic hybrid porous materials. Porous materials usually have the characteristics of a relatively large, specific surface area, high pore melting, easy mass transfer, and high adsorption capacity, so they are widely used in sample pretreatment [76,77].

#### 2.2.1. Synthesis of MOF and Its Application in Sample Pretreatment

In 1995, metal-organic frameworks (MOFs) were created by Yaghi and co-authors [78]. MOF is a new type of organic–inorganic hybrid crystalline porous material with a regular pore or hole structure, which is formed by the connection between an inorganic metal and an organic functional group through a coordination bond, hydrogen bond, π bond, and so on [79]. MOFs have the advantages of regular pore size, large adsorption capacity, high mechanical strength, wide range of morphology, and easy-to-modify properties [80,81]. Researchers can design the pore size and space layout of MOFs according to their needs, and then make specific modifications [82,83,84,85,86]. 

(1) The main preparation method of MOFs in laboratories is the solvothermal method. The principle of a solvothermal method is that inorganic salts and organic linkers are mixed, transferred to a sealed reaction container, and then heated to make the insoluble frameworks grow. Although the solvothermal method is used in the laboratory, it is not suitable for large-scale preparation, because the ratio of surface area to volume is greatly reduced by increasing the volume of the reaction vessel, which affects the nucleation effect of MOFs on the surface of reaction vessel. Moreover, the reaction time of solvothermal method is longer. 

(2) Microwave-assisted solvothermal synthesis was first proposed by Ni and co-workers [87] in 2006. The study showed that microwave-assisted solvothermal synthesis has advantages of fast synthesis speed and good control of crystal size and shape, but it also has the disadvantage of not producing large crystals. 

In 2020, a new type of mixed-matrix membrane (MMM) (Figure 3), based on cationic metal organic frameworks, namely, UiO-66-NMe_3_^+^MMM (Figure 4), was synthesized by Wu and co-workers [88]. It was used to adsorb six kinds of phenoxy carboxylic acids (PCAs) from water prior to ultra-high performance liquid chromatography-tandem mass spectrometry (UHPLC-MS/MS) analysis. Under optimal conditions, UiO-66-NMe_3_^+^MMM showed excellent adsorption efficiency on PCAs. The limit of detection could be as low as 0.03~0.59 ng/L, and the intra-day and inter-day RSD were 2.30% and 3.26%. The results showed that it could be used for the analysis of PCAs in complex water samples such as sewage and reservoir water.

The pyridine triazole functionalized UIO-66 (UiO-66-PYTA) using UiO-66-NH_2_ as a raw material was synthesized by Daliran and co-workers in 2020 [89] (Figure 5). The adsorption of Pd (II) was carried out using UiO-66-PYTA as an adsorbent. Under optimal conditions, UiO-66-PYTA had a good adsorption capacity for Pd (II), and the satisfactory limit of detection (1.9 μg/L) was obtained. The intra-day and inter-day precision are 3.6% and 1.7%. After the reuse of five cycles, the adsorption performance still had no obvious change. More interestingly, the UiO-66-PYTA that adsorbed Pd became an efficient and reusable catalyst for the Suzuki–Miyaura cross-coupling reaction. This kind of adsorption material had its application value both before and after adsorption, and successfully turns waste into treasure. It was consistent with the resource reuse advocated at present.

In 2021, a new type of nano-adsorbent, electrospun, polyacrylonitrile/nickel-based metal-organic framework (PAN/Ni-MOF) nanocomposite coating was synthesized by Amini and co-workers [90] (Figure 6). They added Ni-MOF nanoparticles to PAN nanofibers by one-step method, which improved the porosity of the adsorbents and increased the interaction between adsorbents and analytes via π–π stacking, hydrophobic contact, and hydrogen bonding. The synthesized nanofiber coating had a uniform morphology and porous structure. As a result, the adsorption performance of PAN/Ni-MOF was greatly increased. PAN/Ni-MOF was used to detect diazinon (DIZ) and chlorpyrifos (CPs). It showed an excellent adsorption efficiency for DIZ and CPs, and the limit of detections were 0.3 ng/mL and 0.2 ng/mL. In addition, no harmful organic solvents were added in the process of synthesizing PAN/Ni-MOF, which was in line with the current theme of environmental protection.

#### 2.2.2. Synthesis of a Covalent Organic Framework (COF) and Its Application in Sample Pretreatment

A COF is a new type of porous crystalline organic polymer formed by the polymerization of organic monomers by strong organic covalent bonds. In 2005, the crystalline porous organic polymer was synthesized by Yaghi and co-workers [91], which is the earliest COF, and then in 2007, Yaghi and co-workers [92] synthesized the first three-dimensional COF material. A COF has the advantages of large specific surface area, good stability, low density, high permanent porosity and convenient functionalization. Moreover, the structure can be adjusted, and the aperture, spatial arrangement, and functional properties of the COF can be designed according to the requirements. Functional properties can predict the design so that the COF has a very strong selectivity in the field of separation, and the large specific surface area makes the COF have a larger adsorption capacity. The synthesis of COF materials is difficult, as the construction and ordering of strong covalent bonds are problems that need to be solved. It is one of the reasons that COF materials are not widely used at present; however, with continuous study, solvothermal synthesis and ion thermal synthesis are widely used for the preparation of COF materials. Solvothermal synthesis is a commonly used method for COF synthesis. At present, most COF materials are synthesized by this method, and its specific principle is to change the molding structure of COF by adjusting the pressure, temperature, solvent ratio, and pH in the closed environment. Moreover, the ionic thermal synthesis method uses molten zinc chloride as the solvent and catalyst to make COF crystallize at high temperature. 

In 2019, novel nano-titania modified COFs (NTM-COFs) modified by nano-titanium dioxide was synthesized by Zhao and co-workers [93]. The NTM-COFs were characterized by scanning electron microscopy and transmission electron microscopy, and are shown in Figure 7. NTM-COFs were used as adsorbents in the PRiME pass-through cleanup procedure for the cleanup of local anesthetic drugs in human plasma prior to liquid chromatography-tandem quadrupole mass spectrometry (LC-MS/MS) analysis. The limit of the detections obtained were less than 0.03 μg/L, and the recovery was between 88.8% and 103%. The precision and accuracy of this method were satisfactory. 

In 2020, a novel covalent organic frameworks COF-SCU1 incorporated electrospun nanofibers (PAN@COF-SCU1 nanofibers) (Figure 8) were prepared by Wang and co-workers [94], which were used as adsorbents in pipette tip solid-phase extraction (PT-SPE) (Figure 9) for the adsorption of tetracycline antibiotics (TCs) in food prior to HPLC analysis. The prepared PAN@COF-SCU1 nanofibers showed the characteristics of electrospun nanofibers and COF-Cu1. Under the optimized conditions, PAN@COF-SCU1 nanofibers was used for extraction of tetracycline in grass carp and ducks, and then detected by PT-SPE/HPLC. The limit of detection obtained was 0.6~3 ng/mL, and inter-day and intra-day precision were less than 9.0%. The results showed that PAN@COF-SCU1 nanofibers had excellent adsorption efficiency for TCs, and the established PT-SPE/HPLC method had high precision in the determination of TCs.

In 2020, Zhu and co-workers [95] synthesized novel, petal-shaped, ionic liquid modified COF (PS-IL-COFs) particles using ionic liquid as modifier (Figure 10). They also proposed a new one step cleanup and extraction (OSCE) technique. The OSCE procedure can effectively avoid the problem of the large amount of solvent used, and the long time in liquid–liquid extraction, and can also effectively avoid the cartridge conditioning and eluting steps in SPE. Based on the advantages of this procedure, PS-IL-COFs were used as adsorbents in OSCE for the cleanup of general anesthetics in human plasma prior to the LC-MS/MS analysis. The limit of detections obtained were between 0.0048 µg/L and 0.054 µg/L, and the recovery rate was 82.5–115%.

#### 2.2.3. Synthesis of a Molecularly Imprinted Polymer (MIP) and Its Application in Sample Pretreatment

The concept of molecular imprinting was first proposed by Polykov and co-workers [96]. MIP is a new type of polymer with molecular recognition ability, which has unique predetermination, recognition and practicability [97]. Through the template molecule, the functional monomer and the cross-linking agent can be copolymerized, then the template molecule can be removed, and so the MIP was obtained [98]. It can recognize and adsorb compounds similar to template molecules in shape, size, and chemical function [99,100]. As MIP has strong recognition function, its selectivity is very high, and MIP also has good reusability; therefore, MIP has received increasing attention [101], especially in the field of sample pretreatment. Due to its high selectivity and loading capacity, MIP can be used as an adsorbent in sample pretreatment for the extraction/pre-concentration of trace chemicals. In addition, it has the advantages of low cost, easy preparation, reusability, and high stability in various pH values and temperatures [102,103]. The preparation methods of MIP include suspension polymerization, seed polymerization, emulsion polymerization, and so on [104,105].

Suspension polymerization is the traditional polymerization method, which uses water as a continuous phase to suspend droplets in the presence of stabilizers or surfactants, and then polymerization occurs; however, the particle size of MIP prepared by suspension polymerization is uneven, and the recognition ability is poor. It may be argued that the hydrogen bond and electrostatic interaction in water has an effect on the polymerization process. Seed polymerization is a multi-step swelling polymerization method, in which monodisperse MIP are prepared and then modified in situ; however, water is also used as the continuous phase in this method, so the reaction process may be affected by the hydrogen bond and electrostatic interaction in water. As for emulsion polymerization, it has always been considered as an effective method for the production of highly efficient and monodisperse polymerized particles, and it has been proven that it can be used for the preparation of MIP.

In 2021, a method for the extraction of rosmarinic acid (RA) using MIP was reported by Saad and co-workers [106]. First, the surface imprinted polymer was prepared by using RA as a template, and then it was used as adsorbent for SPE procedure. With the optimized MIP-SPE method (Figure 11), the recovery rate of rosemary can be stabilized at 81.96 ± 6.33%. It can be seen that MIP had excellent adsorption rate on the adsorption of rosemary, and the MIP-SPE method also had further application value in sample pretreatment.

In 2021, a novel, reversible, addition–fragmentation chain transfer polymerization method for the preparation of MIP was reported by Li and co-workers [107]. It was used as a SPE adsorbent for the extraction of four phenylarsine compounds in feed, edible chicken, and pork samples. The results showed that the recovery rate of phenylarsine compounds by the developed MIP-SPE method was between 83.4% and 95.1%.

#### 2.2.4. Synthesis of Porous Hybrid Material and Its Application in Sample Pre-Treatment

With the continuous improvement to the requirements of analytical accuracy and the increasing complexity of analytical targets, the performance of a single material is gradually unable to meet the current needs, so researchers focus on hybrid materials, among which, the hybrid of porous materials is one of the important research directions. Graphene oxide is one of the more popular materials in hybrid materials, because graphene oxide is rich in oxygen-containing groups, and has a π stacking interaction, thermal stability, mechanical stability, and high specific surface area. 

In 2022, a novel Cu-based metal–organic framework/graphene oxide (Cu–MOF/GO) hybrid nanocomposite coated on a stainless-steel mesh using the sol–gel method was prepared by Abdar and co-workers [108]. The hybrid nanocomposite was used as semi-automatic SPE sorbent for the extraction of polycyclic aromatic hydrocarbons. The extracted PAHs were analyzed by gas chromatography-flame ionization detection (GC-FID). Under the optimal experimental conditions, the detection limit was 0.3–1.8 pg/mL, the relative recovery was between 95.1–99.5%, and the relative standard deviation (RSD%) was between 4.9–6.9%. This method combined the advantages of GO, Cu-MOF, and the sol–gel method, and thus, this hybrid nanomaterial had a variety of synergistic effects.

Moreover, an effective adsorbent of polypyrrole doped GO/COF-300 (GO/COF-300/PPy) nanocomposite was prepared for the first time by Feng and co-workers in 2022 [109]. This hybrid material was used for the adsorption of indomethacin (IDM) and diclofenac (DCF) in aqueous solution. The concentration of IDM and DCF was determined by a UV-vis spectrometer at 264 nm and 274 nm. The results showed that the adsorption efficiency of GO/COF-300/PPy composite for IDM and DCF was much higher than that of single GO and COF, and the adsorption capacity of DIM and DCF were 115 mg/g and 138 mg/g, respectively. And after 8 times of adsorption-desorption process, the adsorption efficiency of GO/COF-300/PPy composite materials for IDM and DCF decreased by 27% and 29%, indicating that GO/COF-300/PPy has good reusability and stability.

### 2.3. Magnetic Nanomaterials

As we know, the common ferromagnetic materials in nature are Fe_2_O_3_, Fe_3_O_4_, ferrites (MFe_2_O_4_, M=Mn, Zn, Co, Ni, Cu) and so on. It has become a trend of scientific research to magnetize nano-materials and improve their properties. In recent years, many magnetic nanomaterials have been reported, such as γ-Fe_2_O_3_, Fe_3_O_4_, Mn_3_O_4_, MnO, CoFe_2_O_4_, MnFe_2_O_4_. The preparation methods of magnetic nanomaterials are mainly divided into three categories; (1) chemical synthesis methods include aqueous solution coprecipitation method, hydrothermal method, microemulsion method and thermal decomposition method; (2) physical synthesis methods include mechanical ball milling method and evaporation condensation method; (3) the biosynthesis method.

Magnetic nanocomposites have different properties from traditional materials, such as superparamagnetism, high coercivity, low curie temperature, and high susceptibility. With the application of magnetic field, the magnetic adsorbent can be easily separated from the matrix solution after adsorbing the target analyte; therefore, there is no need for centrifugation or filtration. Recently, MSPE has been developed rapidly. MSPE uses magnetic materials as solid-phase adsorbents, which can greatly simplify the SPE procedure and improve the extraction efficiency. 

Although magnetic nanomaterials are widely used as adsorbents in MSPE, magnetic nanoparticles are prone to agglomeration due to magnetic nanometer size, large specific surface area, high surface energy and reaction activity. In order to increase the structural stability and enhance the specific binding between magnetic nanoparticles and the target analytes, the surface of magnetic particles should be functionalized. After the surface modification of magnetic particles by different methods, magnetic nanoparticles can effectively reduce the phenomenon of aggregation, enhance the stability in solution. 

(1) Modification with polymer: magnetic nanoparticles have poor stability and biocompatibility, and are thus easy to be oxidized. The surface of magnetic nanoparticles can be modified by polymer materials with good biocompatibility. In most cases, the magnetic nanocomposites synthesized are core-shell structures. There are two ways to do this for magnetic nanoparticles coated with polymer. One is physical coating, that is, polymers are directly coated on the surface of magnetic nanoparticles to synthesize organic polymers in situ. Another method is chemical coating, which is to modify the surface of magnetic nanoparticles to connect their surfaces with specific functional groups. Then the functional groups can react with polymers to form magnetic nanocomposites.

A new modification method for the preparation of N-dopedmagnetic covalent organic framework (N-Mag-COF) was reported by Wu and co-workers [110]. It has been evaluated in the magnetic dispersive solid phase extraction (Mag-dSPE) procedure for allergenic disperse dyes prior to LC-MS/MS analysis. Under optimal conditions, the N-Mag-COF Mag-dSPE procedure can significantly reduce the matrix effect and obtain satisfactory recoveries with RSD lower than 8.0%

A method for the synthesis of novel magnetic nanocomposites was reported by Lu and co-workers [111]. Fe_3_O_4_ particles functionalized by polydopamine was grafted onto Zr-MOF by layer modification, and then the hydrophobic carboxyl functionalized ionic liquids (IL-COOH) were coated onto the Fe_3_O_4_@Zr-MOF magnetic particles, and the novel IL-COOH/Fe_3_O_4_@Zr-MOF magnetic nanocomposites were obtained (Figure 12). IL-COOH/Fe_3_O_4_@Zr-MOF had been used as an adsorbent for the extraction of fluoroquinolone antibiotics in environmental water samples prior to HPLC analysis. The results showed that IL-COOH/Fe_3_O_4_@Zr-MOF had an excellent adsorption efficiency for fluoroquinolones antibiotics, and the maximum adsorption capacity of ofloxacin was 438.5 mg/g, the recovery of ofloxacin in environmental water samples was between 90.0% and 110.0%, and the limit of detection was less than 0.02 μg/L. 

Li and co-workers [112] reported a green method for the preparation of hydrophilic magnetic MIP composites for the extraction of cyclic adenosine monophosphate (cAMP). Magnetic carbon nanotubes (MCNTs) modified by tetraethyl orthosilicate (TEOS) was embedded into the chitosan (CS) network to obtain cAMP-MIPs@MCNTs composites (Figure 13 and Figure 14). The cAMP-MIPs@MCNTs had been used as adsorbents to extract cAMP from winter prior to HPLC analysis. The results showed that cAMP-MIPs@MCNTs have a high adsorption capacity for cAMP. The limit of detection was 5 ng/mg, and the imprinting factor was 2.94, which indicated that cAMP-MIPs@MCNTs had excellent selectivity for the template molecules.

(2) Modification with carbon materials: carbon materials have high thermal and chemical stability and good biocompatibility. Magnetic nanoparticles coated with carbon materials can not only prevent the agglomeration of nanoparticles, but also prevent oxidation corrosion and enhance their stability. 

In 2018, a novel double-layered pipette tip magnetic dispersive solid phase extraction (DPT-MSPE), based on polyamide functionalized magnetic carbon nanotubes (PAMAM@Mag-CNTs) (Figure 15), was synthesized by Cheng and co-workers [113]. It was applied to the adsorption of fifteen toxic alkaloids in vegetables and meat. Under the optimal conditions, the satisfactory recoveries were between 83.4% and 125%, and the RSD was less than 8%. 

In 2018, an enhanced cleanup efficiency hydroxy functionalized-magnetic GO (EH-MAG-GO) modified by hydroxyl group was successfully synthesized by Ma and co-workers (Figure 16 and Figure 17) [9]. EH-MAG-GO was used as an adsorbent in the PRiME pass-through cleanup procedure for the cleanup of strychnine and brucine prior to LC-MS/MS analysis. Under the optimized conditions, the limit of detections for strychnine and brucine were 0.088 μg/L and 0.092 μg/L, and the recovery was 89.4–118%. Notably, the EH-Mag-GO can be reused (20 times) without much sacrifice of cleanup efficiency, and the validation results demonstrated the applicability of the EH-MAG-GO PRiME pass-through cleanup procedure for clinical studies.

GO-Fe_3_O_4_ magnetic nanocomposites, using a simple chemical coprecipitation method, was successfully synthesized by Lu and co-workers [114]. In this method, Fe_3_O_4_ formed by the co-precipitation of Fe^2+^ and Fe^3+^ was successfully bonded to the GO with the aid of the -COOH under ultrasound. The preparation process of GO-Fe_3_O_4_ was shown in Figure 18. GO-Fe_3_O_4_ was applied to SPME as an adsorbent to extract eight psychoactive drugs from urine prior to LC-MS/MS analysis. The limit of detection of the analytical method was 0.02–0.2 μg/L, and the intra-day and inter-day RSD were in the range of 2.7–13.1% and 3.9–13.7%, respectively. 

Furthermore, the performance of nanomaterials as adsorbents in combination with other technologies have been summarized, and the results are shown in Table 1. As shown in the table, the proposed sample pretreatment with nanomaterials as adsorbents give much lower LODs of various chemical analytes.

### 2.4. Regeneration and Reproducibility of Nanomaterials

When nanomaterials are used as adsorbents, the regeneration and reproducibility of the adsorbents are very important evaluation indicators. 

In order to achieve good regeneration, the selection of eluent is very important, which is also the key to the optimization of the solid phase extraction process. As for the reproducibility of the nanomaterial, it is necessary to carry out parallel experiments, and then calculate the relative deviation of these parallel experiments. For example, Zhao and co-workers [115] reported a novel, three-dimensional, interconnected, magnetic, chemically modified graphene oxide (3D-Mag-CMGO) for the enrichment of dispersed dyes in water samples. The 3D-Mag-CMGO-adsorbed could be conveniently regenerated by using 5.0% ammonia solution/methanol (*v*/*v*) and water as an eluent, allowing for them to be used repeatedly for Mag-dSPE method. After adsorption–desorption experiments, the results showed that 3D-Mag-CMGO could be reused at least ten times without much sacrifice of the extraction efficiency. In addition, six parallel experiments were carried out, and the relative deviation of the recovery rate of the target substance were between 4.6–8.0%. In 2018 [116], they tested the regeneration and reproducibility of quaternary ammonium modified magnetic carboxyl-carbon nanotubes (QA-Mag-CCNTs) and found that ammonia solution/water (5.0% *v*/*v*) was the best eluent. After desorption, the recovery rate of the target substance could reach 89.2%. Moreover, after 20 times of adsorption–desorption experiments, the adsorption efficiency of the nanomaterial to the target did not decrease significantly. Furthermore, after eight parallel experiments, the relative standard deviation was between 3.6–7.0%, and in the same year, Zhang and co-workers tested the regeneration and reproducibility of the three-dimensional ionic liquid-ferrite functionalized graphene oxide nanocomposite (3D-IL-Fe_3_O_4_-GO) [117]. For optimal regeneration, they conducted a lot of experiments. Results showed that the adsorbed 3D-IL-Fe_3_O_4_-GO could be conveniently regenerated by washing with toluene and water sequentially. In addition, 3D-IL-Fe_3_O_4_-GO could be reused at least 10 times with nearly invariable quantitative efficiency. For reproducible detection, five batches of 3D-IL-Fe_3_O_4_-GO were prepared and applied in five parallel sets of experiments, the relative deviation is between 4.5% and 8.6%.

## 3. Conclusions and Outlook

In this review, we briefly summarize the latest progress in the preparation of nanomaterials and their applications in sample pretreatment methods. Numerous selected nanomaterials have been described in detail. Combined with these studies, it can be proved that the application of nanomaterials in sample pretreatment has great potential in the future. Although the adsorption efficiency, selectivity, and reproducibility of nanomaterials are better than those of traditional adsorbents, there are still some problems and challenges in nanomaterials that need to be overcome.

(1) The structure of carbon nanomaterials and magnetic nanomaterials are relatively simple, and the selectivity is poor, which greatly affects its practical application. In order to improve the selectivity of carbon nanomaterials and magnetic nanomaterials, multi-modification of specific functional groups should be considered.

(2) The synthesis method of MOFs’ and COFs’ nanomaterials are complex, and some expensive or toxic reagents may be used in the synthesis process. Reducing the synthesis complexity of existing nanomaterials and increasing its reproducibility and repeatability are key factors in the future study.

(3) For the MIP nanomaterials, it takes long incubation times to induce the binding of the analyte onto the recognition sites on the MIP surface, and these recognition sites will also bind to some molecules with similar structure to the target analyte, thus affecting the selectivity of MIP nanomaterials. In addition, the non-compound forms of MIP are very limited studies at present, which limits its scope of application. The research on the non-complex form of MIP should be strengthened to further understand the adsorption mechanism of MIP, so as to find ways to improve its selective ability to identify target analytes.

## Figures and Tables

**Figure 1 nanomaterials-12-01845-f001:**
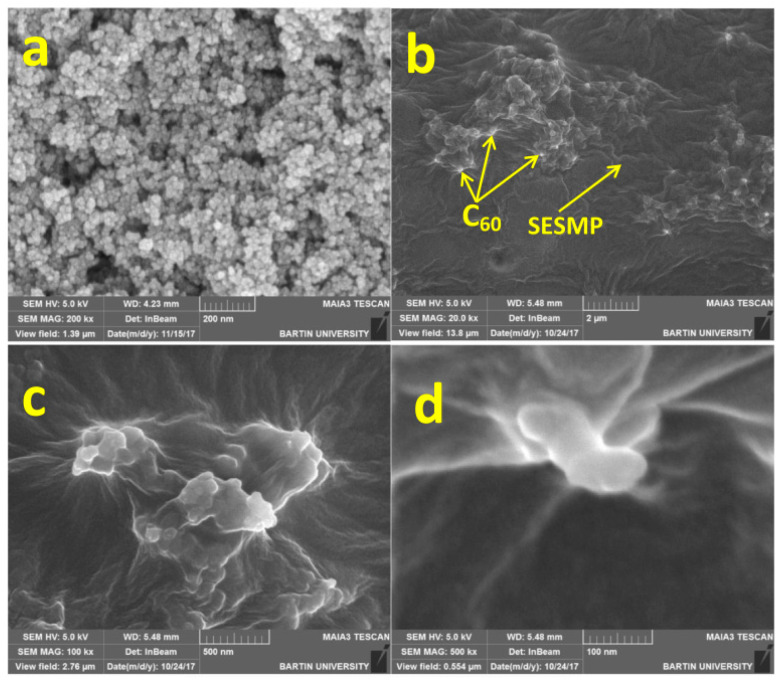
SEM of (**a**) Fe_3_O_4_-NPs, (**b**) C_60_-SESMP nanocomposite 2 μm, (**c**) C_60_-SESMP nanocomposite 500 nm, and (**d**) C_60_-SESMP nanocomposite 100 nm. Reprinted with permission from Ref. [48]. Copyright (2019) Elsevier.

**Figure 2 nanomaterials-12-01845-f002:**
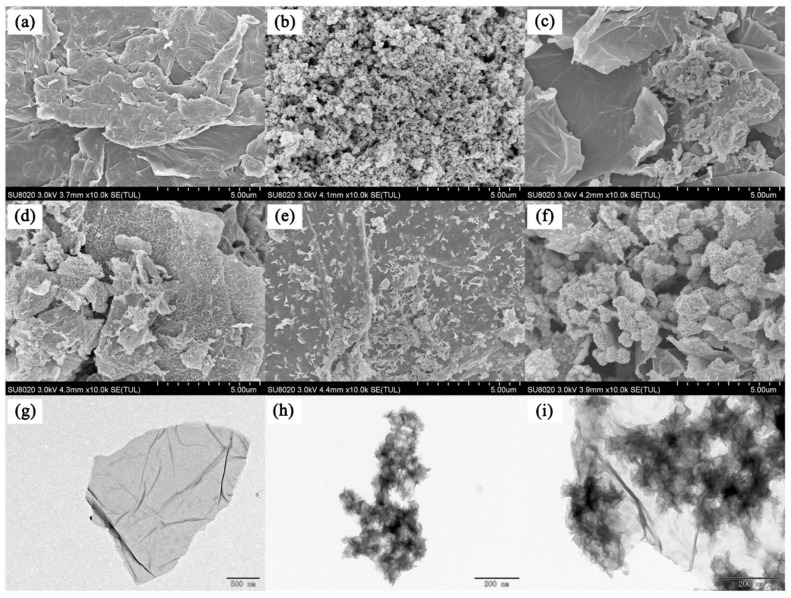
SEM images of GO: (**a**) MoS_2_, (**b**) MoS_2_/GO-0.5, (**c**) MoS_2_/GO-1, (**d**) MoS_2_/GO-1.5, (**e**) andMoS_2_/GO-2. (**f**) TEM images of GO: (**g**) MoS_2_ (**h**) and MoS_2_/GO-1.5. (**i**). Reprinted with permission from Ref. [71]. Copyright (2021) Springer Wine.

**Figure 3 nanomaterials-12-01845-f003:**
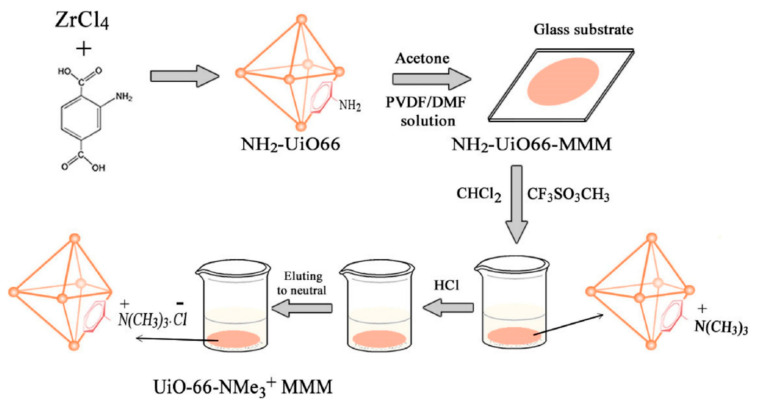
Illustration for the synthesis of the UiO-66-NMe_3_^+^ MMM. Reprinted with permission from Ref. [88]. Copyright (2020) Elsevier.

**Figure 4 nanomaterials-12-01845-f004:**
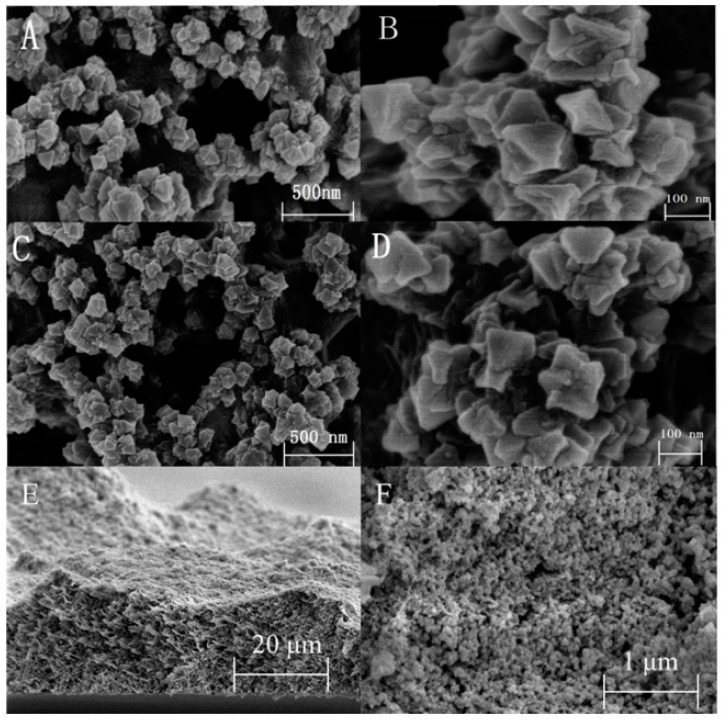
SEM images of UiO-66-NH_2_ MMM with the scale bar of (**A**) 500 nm and (**B**) 100 nm. (**C**) UiO-66-NMe_3_^+^ MMM with the scale bar of 500 nm and (**D**) 100 nm. (**E**) Cross-sectional SEM images of UiO-66-NMe_3_^+^ MMM with the scale bar of 20 μm and (**F**) 1 μm. Reprinted with permission from Ref. [88]. Copyright (2020) Elsevier.

**Figure 5 nanomaterials-12-01845-f005:**
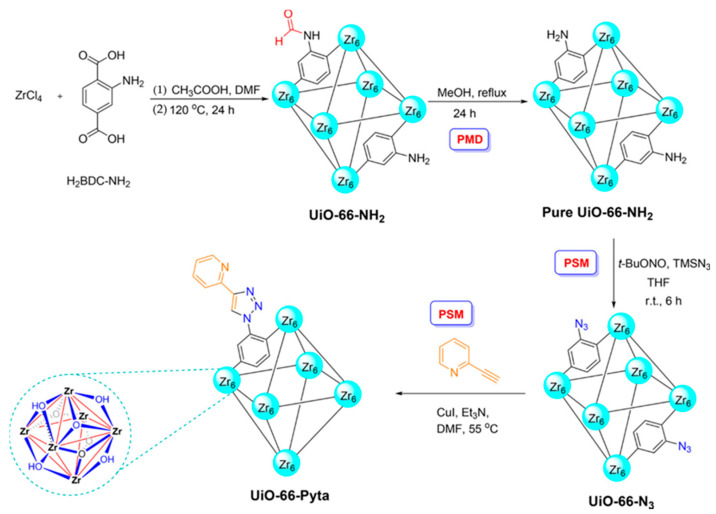
Schematic Synthesis: PMD and PSMs of UiO-66-NH2 toward formation of UiO-66-Pyta. Reprinted with permission from Ref. [89]. Copyright (2020) American Chemical Society.

**Figure 6 nanomaterials-12-01845-f006:**
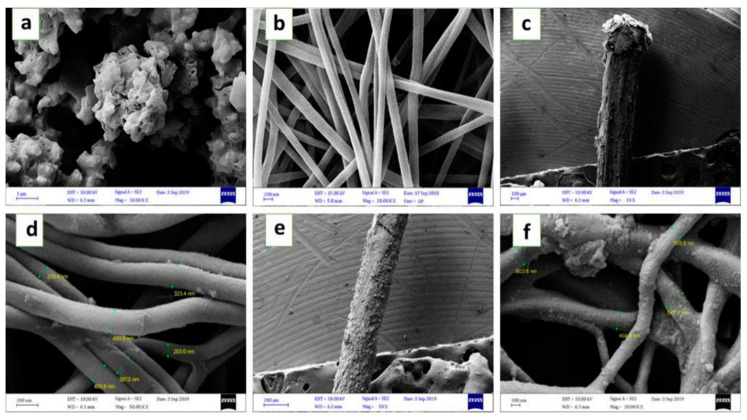
FESEM images of (**a**) Ni-MOF particles synthesized by hydrothermal method; (**b**) PAN nanofibers; (**c**,**d**) PAN/Ni-MOF composite nanofibers before extraction in different magnification; (**e**,**f**) electrospun PAN/Ni-MOF nanofibers after 160 times extraction in different magnification. Reprinted with permission from Ref. [90]. Copyright (2021) Elsevier.

**Figure 7 nanomaterials-12-01845-f007:**
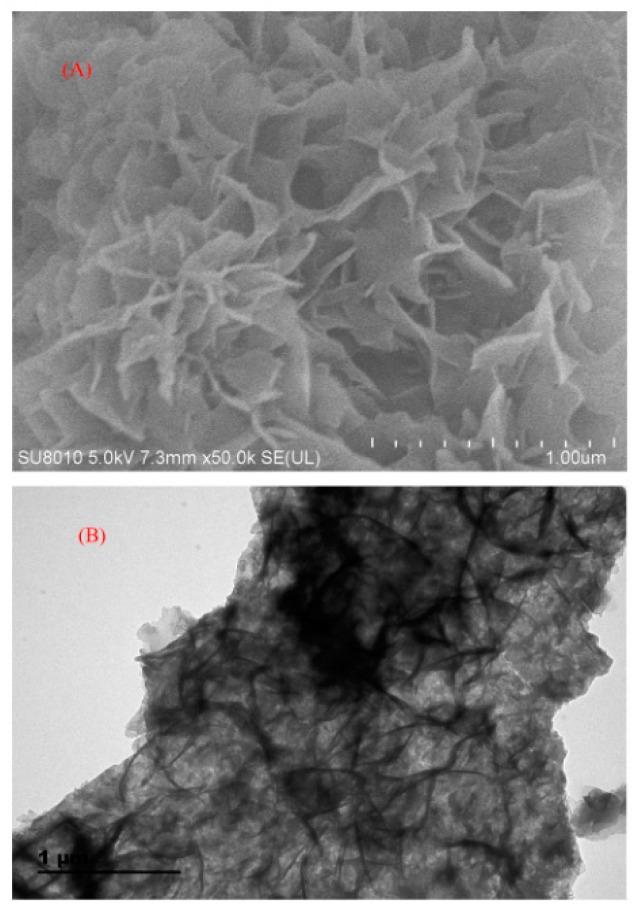
SEM (**A**) and TEM (**B**) images of NTM-COFs. Reprinted with permission from Ref. [93]. Copyright (2019) Elsevier.

**Figure 8 nanomaterials-12-01845-f008:**
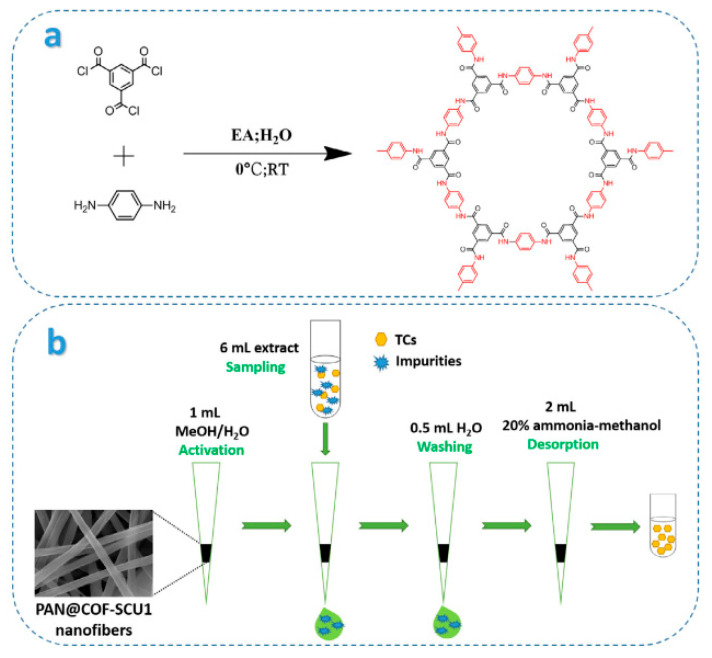
Schematic diagrams of (**a**) preparation of COF-SCU1 and (**b**) PT-SPE procedure for the extraction of TCs. Reprinted with permission from Ref. [94]. Copyright (2020) Elsevier.

**Figure 9 nanomaterials-12-01845-f009:**
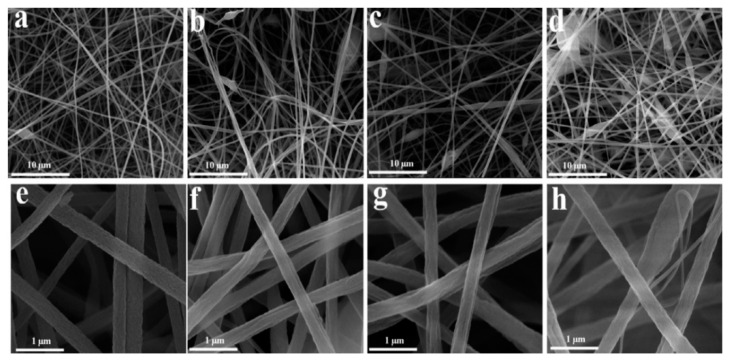
(**a**,**e**) SEM images of PAN nanofibers; (**b**,**f**) PAN@COF-SCU1 (0.1); (**c**,**g**) PAN@COF-SCU1 (0.2); (**d**,**h**) PAN@COF-SCU1 (0.3). Reprinted with permission from Ref. [94]. Copyright (2020) Elsevier.

**Figure 10 nanomaterials-12-01845-f010:**
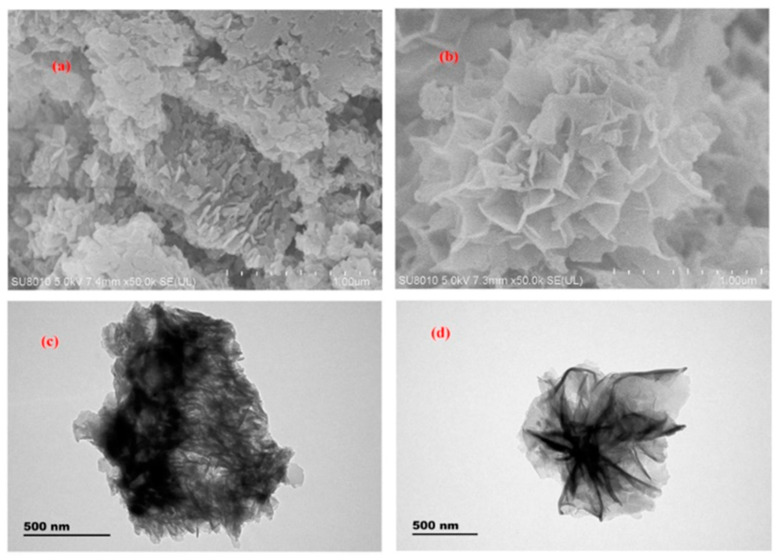
(**a**) SEM images of COFs and (**b**) PS-IL-COFs. (**c**) TEM images of COFs and (**d**) PS-IL-COFs. Reprinted with permission from Ref. [95]. Copyright (2020) Elsevier.

**Figure 11 nanomaterials-12-01845-f011:**
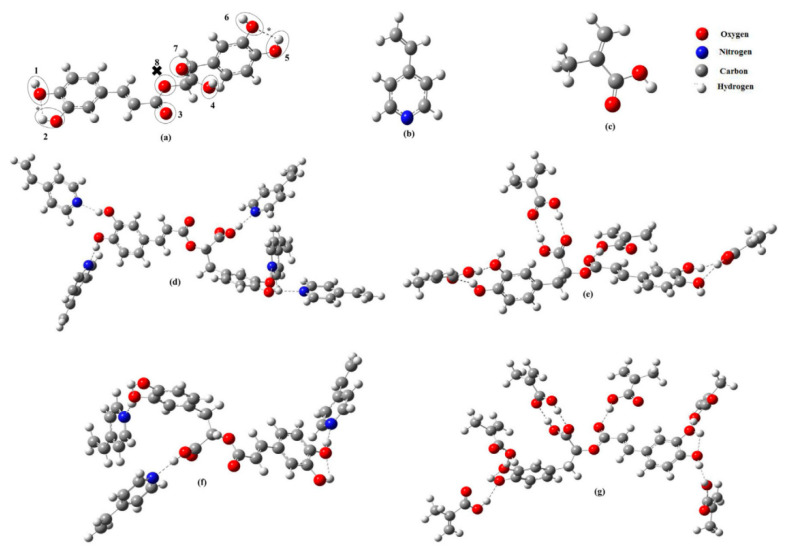
Computer optimized structures of (**a**) RA, (**b**) 4-VP, and (**c**) MAA, conformations, optimum. (**d**) RA-(4-VP)_5_ and (**e**) RA-(MAA)_4_ conformations assuming no intra-molecular H-bond formation, optimum (**f**) RA-(4-VP)_3_, and (**g**) RA-(MAA)_6_ conformations assuming intra-molecular H-bond formation. The possible active binding sites are encircled, * represents possible intra-molecular H-bonds of a distance (2.166–2.174 Ǻ), and x represents the site incapable of H-bonding with MAA. Reprinted with permission from Ref. [106]. Copyright (2021) Elsevier.

**Figure 12 nanomaterials-12-01845-f012:**
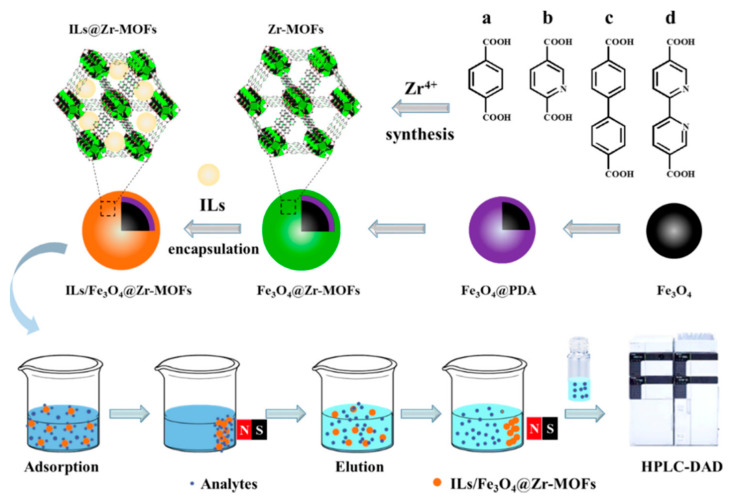
Preparation of ILs/Fe_3_O_4_@Zr-MOFs and MSPE Process. Reprinted with permission from Ref. [111]. Copyright (2021) American Chemical Society.

**Figure 13 nanomaterials-12-01845-f013:**
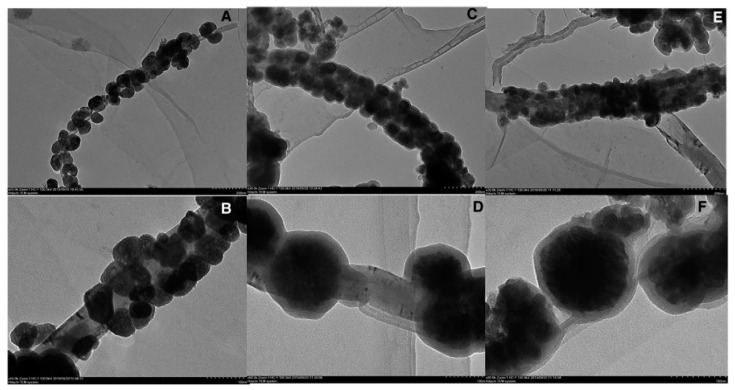
(**A**,**B**) TEM images of the prepared MCNTs, (**C**,**D**) TEOS@MCNTs, and (**E**,**F**) cAMP-MIPs@MCNTs. Reprinted with permission from Ref. [112]. Copyright (2021) Weinheim.

**Figure 14 nanomaterials-12-01845-f014:**
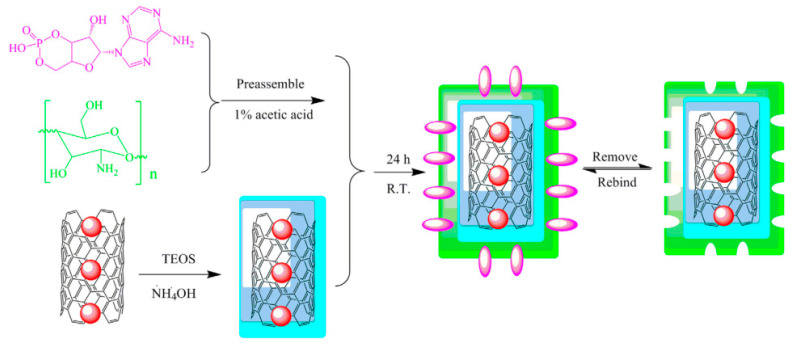
The synthesis procedure of cAMP-MIPs@MCNTs. Reprinted with permission from Ref. [112]. Copyright (2021) Weinheim.

**Figure 15 nanomaterials-12-01845-f015:**
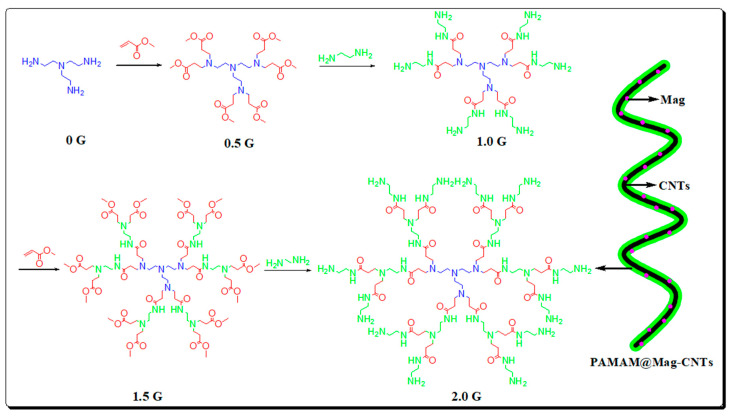
The synthetic process of PAMAM. Reprinted with permission from Ref. [113]. Copyright (2018) Royal Society Chemistry.

**Figure 16 nanomaterials-12-01845-f016:**
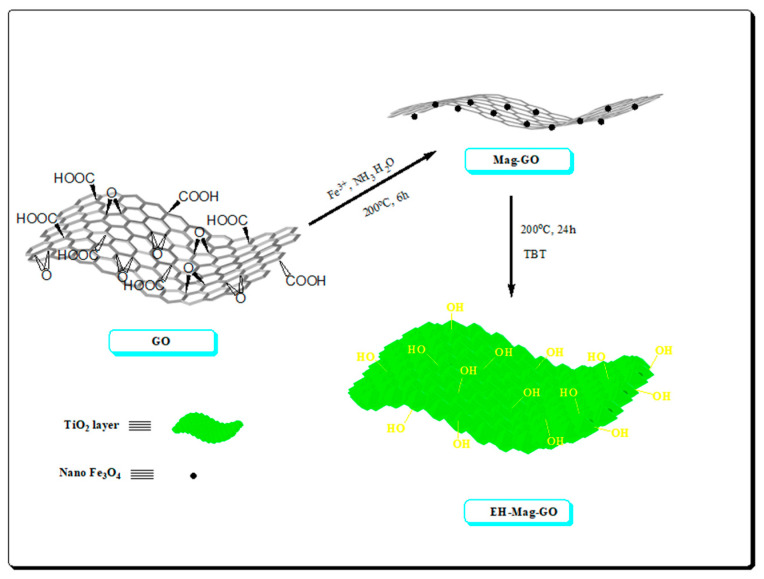
The preparation procedure of EH-Mag-GO. Reprinted with permission from Ref. [9]. Copyright (2018) Elsevier.

**Figure 17 nanomaterials-12-01845-f017:**
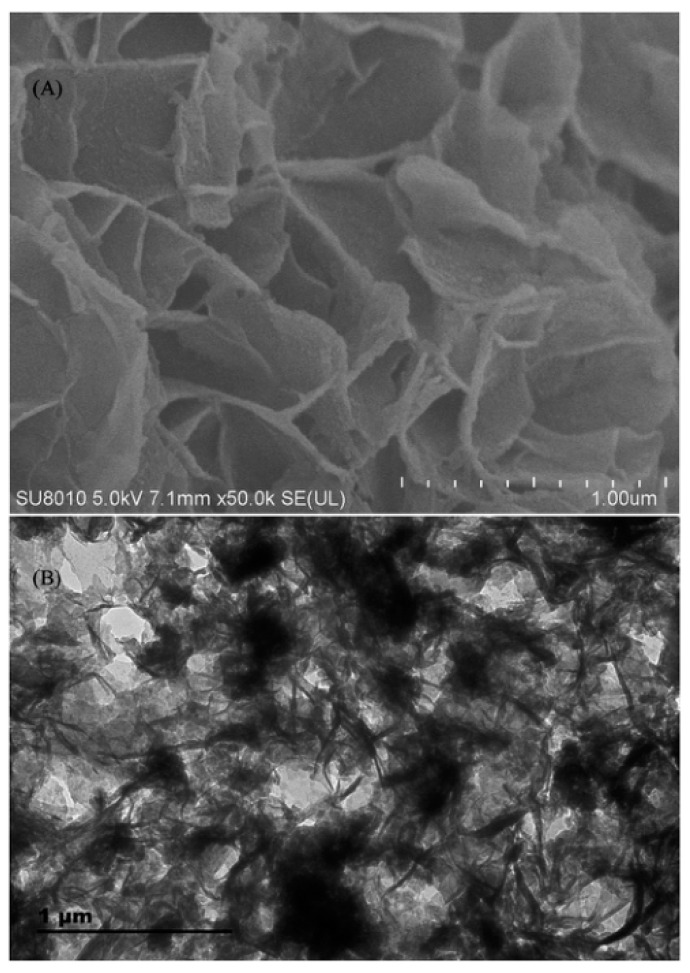
SEM (**A**) and TEM (**B**) images of EH-Mag-GO. Reprinted with permission from Ref. [9]. Copyright (2018) Elsevier.

**Figure 18 nanomaterials-12-01845-f018:**
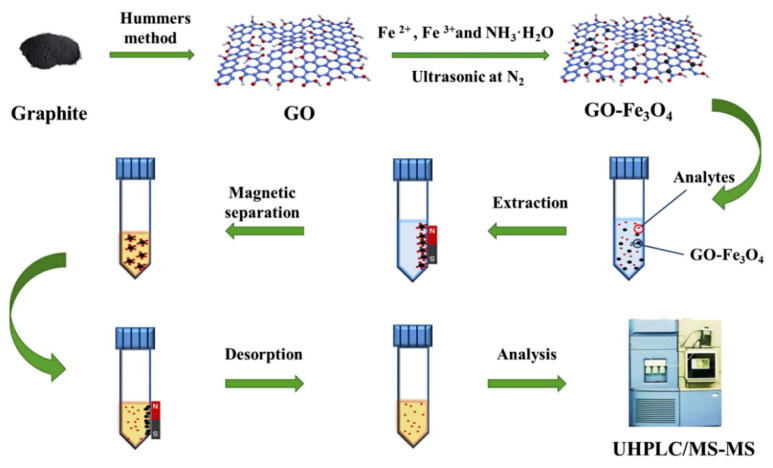
Schematic illustration of the synthesis of GO–Fe_3_O_4_ and MSPE procedure for analysis of the eight psychoactive drugs. Reprinted with permission from Ref. [114]. Copyright (2020) Elsevier.

**Table 1 nanomaterials-12-01845-t001:** The summary of the properties of nanomaterials as adsorbents in combination with other technologies.

Sample	Target Analyte	Adsorbent	Pretreatment Method	Detection	LOD (µg/mL)	Recovery (%)	Ref.
Crude oil	As(III) As(V)	C60-SESMP-Fe_3_O_4_	MSPE	ICP-OES	0.047–0.032	96.5–98.0	[48]
Wild rice	Pb (II)	ox-MWCNTs/ batophenanth	DMSPE	FAAS	0.25	93.8	[60]
Sewage water	U(VI)	Br-PADAP/MWCNT	MSPE	ICP-AES	0.14	99.7	[61]
Cosmetics	Parabens	MoS_2_/GO	DSPE	UPLC-PDA	0.40~2.3	91.3–124	[71]
Sewage water	PCAs	UIO-66-NMe_3_^+^MMM	DME	UHPLC-MS/MS	0.030–0.59	80.1–117	[88]
Tap water	Pd(II)	UIO-66-Pyta	SPE	ICP-AES	1.9	99.0	[89]
Duck, Grass Carp	TCs	PAN@COF-SCU1	PT-SPE	HPLC	0.60–3.0	84.0–117	[94]
Human Plasma prior	Anesthetics	PS-IL-COFS	OSCE	LC-MS/MS	0.016–0.18	82.5–115	[95]
Rice, apples, green groceries	PCAs	TAPT-DHTA-COF	SPE	LC-MS/MS	0.0070–0.030	81.2–107	[96]
River water	OPPS	PAN/Ni-MOF	HS-SPME	CD-IMS	0.20–0.30	87.0–98.0	[90]
River water	FQS	IL-COOH/Fe_3_O_4_@Zr-MOF	MSPE	HPLC-DAD	<0.020	90.0–110	[111]
Meat, vegetables	Toxic alkaloids	PAMAM@Mag-CNTs	DPT-MSPE	UFLC-MS/MS	0.011~0.329	83.4–125	[113]
Urine	Psychoactive drugs	GO-Fe_3_O_4_	MSPE	UHPLC-MS/MS	0.020–0.20	80.4–105	[114]

## Data Availability

Not applicable.

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
