# Peer review of "Nanomaterials with Excellent Adsorption Characteristics for Sample Pretreatment: A Review"

_nanomaterials, 2022, doi:10.3390/nano12111845_

Round 1
Reviewer 1 Report
Comments from Reviewer
Title: Nanomaterials with excellent adsorption characteristics for sample pretreatment: A review
The current form's presentation of methods and scientific results is satisfactory for publication in the Nanomaterials journal. The minor and significant drawbacks to be addressed can be specified as follows:
1. Line 3. and Yong ---> Yong.
2. Line 88, “by Kroto and co-workers in 1985[39];” [39] Allaf, A.W.; Balm, S. C60 Buckminsterfullerene. Nature. 1985, 318, 162-163. ???? Invalid reference.. Please use Kroto, H.W.; Heath, J. R.; Obrien, S. C.; Curl, R. F.; Smalley, R. E. (1985). "C60: Buckminsterfullerene". Nature. 318 (6042): 162–163.
3. Line 163: nanotubes: (1) ---> nanotubes: (i) line 164: (2) there are metal ---> (ii) there are metal.
4. Line 200. -COO- ---> -COOH.
5. Line 91: Novoselov and co-workers [41]. But line 209: Geim and co-workers [41]. ???
6. Line 223. Ilaria Silvestro and co-workers ---> Silvestro and co-workers.
7. Line 244, (…)microporous materials (pore size less than 2 nm) [72-75]. Incorrect references. Please take into account world-known books (Rouquerol, Gogotsi, etc.) What about papers with IUPAC recommendations?
8. Line 300. toward Formation ---> toward formation.
9. Line 381. MIP?
10. Line 420, The possible active binding sites are encircled, * represents.
11. Fig. 12, Background. https:\\pubs.acs.o????
Sincerely,
The reviewer.
Author Response
Dear editor,
We are truly grateful to yours and other reviewers’ critical comments and thoughtful suggestions. Based on these comments and suggestions, we have made careful modifications on the original manuscript. All changes made to the text are in red color. We hope the new manuscript will meet your magazine’s standard. Below you will find our point-by-point responses to the reviewers’ comments/questions:
Comments from Reviewer 1
Title: Nanomaterials with excellent adsorption characteristics for sample pretreatment: A review
The current form's presentation of methods and scientific results is satisfactory for publication in the Nanomaterials journal. The minor and significant drawbacks to be addressed can be specified as follows:
- Line 3. and Yong ---> Yong.
√ Many thanks for your suggestion. We have corrected the mistake. Thank you very much!
- Line 88, “by Kroto and co-workers in 1985[39];” [39] Allaf, A.W.; Balm, S. C60 Buckminsterfullerene. Nature. 1985, 318, 162-163. ???? Invalid reference.. Please use Kroto, H.W.; Heath, J. R.; Obrien, S. C.; Curl, R. F.; Smalley, R. E. (1985). "C60: Buckminsterfullerene". Nature. 318 (6042): 162–163.
√ Many thanks for your suggestion. We have corrected the mistakes. Thank you very much!
- Line 163: nanotubes: (1) ---> nanotubes: (i) line 164: (2) there are metal ---> (ii) there are metal.
√ Many thanks for your suggestion. We have corrected the mistakes. Thank you very much!
- Line 200. -COO- ---> -COOH.
√ Many thanks for your suggestion. We have corrected the mistake. Thank you very much!
- Line 91: Novoselov and co-workers [41]. But line 209: Geim and co-workers [41]. ???
√ Many thanks for your suggestion. In line 209, it should be Novoselov and co-workers [41], and we have corrected the mistake. Thank you very much!
- Line 223. Ilaria Silvestro and co-workers ---> Silvestro and co-workers.
√ Many thanks for your suggestion. We have corrected the mistake. Thank you very much!
- Line 244, (…)microporous materials (pore size less than 2 nm) [72-75]. Incorrect references. Please take into account world-known books (Rouquerol, Gogotsi, etc.) What about papers with IUPAC recommendations?
√ Many thanks for your suggestion. According to your suggestion, we have corrected the related references, and the book (Rouquerol, Gogotsi, etc.) has been added to the manuscript as below, thank you very much!
[75] Rouquerol, F.; Rouquerol, J.; Sing, K.S.W.; Llewellyn, P.; Maurin, G. Adsorption by powders and porous solids: principles, methodology and applications. Academic Press. 1999, 163-165.
- Line 300. toward Formation ---> toward formation.
√ Many thanks for your suggestion. We have corrected the mistake. Thank you very much!
- Line 381. MIP?
√ Many thanks for your suggestion. The “MIP” has been corrected to “molecularly imprinted polymer (MIP)”. Thank you very much!
- Line 420, the possible active binding sites are encircled, * represents.
√ Many thanks for your suggestion. We have corrected the mistakes. Thank you very much!
Figure 11. Computer optimized structures of (a) RA, (b) 4-VP, and (c) MAA, conformations, optimum (d) RA-(4-VP)5 and (e) RA-(MAA)4 conformations assuming no intra-molecular H-bond formation, optimum (f) RA-(4-VP)3 and (g) RA-(MAA)6 conformations assuming intra-molecular H-bond formation. The possible active binding sites are encircled, * represents possible intra-molecular H-bonds of a distance (2.166–2.174 Ǻ), and x represents the site incapable of H-bonding with MAA. Reprinted with permission from Ref. [106]. Copyright (2021) Elsevier.
- Fig. 12, Background. https:\\pubs.acs.o????
√ Many thanks for your suggestion. We have corrected the mistake. Thank you very much!
Figure 12. Preparation of ILs/Fe3O4@Zr-MOFs and MSPE Process. Reprinted with permission from Ref. [111]. Copyright (2021) American Chemical Society.
Yours sincerely
Prof. Yong-Gang Zhao
Zhejiang Shuren University, Hangzhou 310015, China
E-mail address: zhyg91213@163.com

Reviewer 2 Report
Manuscript number: nanomaterials-1729105
Title: “Nanomaterials with excellent adsorption characteristics for sample pretreatment: A review”
Authors: Wen Xin Liu, Shuang Song, Ming Li Ye, Yan Zhu, Yong Gang Zhao, Yin Lu
The paper concerns of nanomaterials with excellent adsorption properties and characteristics for sample pretreatment. The work focuses on the following aspects: carbon nanomaterials, porous nanomaterials and magnetic nanomaterials. The paper brings valuable information and its scope is interesting. However, it contains some arguable elements. A proper explanations are needed. Comments and reservations:
· * The wording heavy metal ions is a imprecise concept. Therefore, next to this phrase, I would also include the name: hazardous metal ions.
* 2.2. Porous nanomaterials - I definitely miss information about porous nanomaterials in this chapter. A wide range of oxide-based (nano-sized) hybrid systems also play a crucial role. I recommend that you include information about this aspect as well.
· * No information available on the desorption / regeneration process of the materials used. I recommend that this be completed.
Final remark: In my opinion, the paper is interesting and is well prepared – especially part concerning Magnetic nanomaterials. In my opinion, the paper is worth recommendation for publication in Materials after minor revision.
Yours faithfully
Author Response
Dear editor,
We are truly grateful to yours and other reviewers’ critical comments and thoughtful suggestions. Based on these comments and suggestions, we have made careful modifications on the original manuscript. All changes made to the text are in red color. We hope the new manuscript will meet your magazine’s standard. Below you will find our point-by-point responses to the reviewers’ comments/questions:
Comments from Reviewer 2
The paper concerns of nanomaterials with excellent adsorption properties and characteristics for sample pretreatment. The work focuses on the following aspects: carbon nanomaterials, porous nanomaterials and magnetic nanomaterials. The paper brings valuable information and its scope is interesting. However, it contains some arguable elements. A proper explanations are needed. Comments and reservations:
- The wording heavy metal ions is a imprecise concept. Therefore, next to this phrase, I would also include the name: hazardous metal ions.
√ Many thanks for your suggestion. According to your suggestion, “heavy metal ions” have been corrected to “hazardous metal ions”. Thank you very much!
- 2.Porous nanomaterials - I definitely miss information about porous nanomaterials in this chapter. A wide range of oxide-based (nano-sized) hybrid systems also play a crucial role. I recommend that you include information about this aspect as well.
√ Many thanks for your suggestion. According to your suggestion, the oxide-based (nano-sized) hybrid porous nanomaterials have been added to the manuscript as below, thank you very much!
“2.2.4 Synthesis of porous hybrid material and its application in sample pretreatment
With the continuous improvement of the requirements of analytical accuracy and the increasing complexity of analytical targets, the performance of a single material is gradually unable to meet the current needs, so researchers focus on hybrid materials, among which the hybrid of porous materials is one of the important research directions. Graphene oxide is one of the more popular materials in hybrid materials, because graphene oxide is rich in oxygen-containing groups, and has π stacking interaction, thermal stability, mechanical stability and high specific surface area.
In 2022, a novel Cu-based metal-organic framework/graphene oxide (Cu-MOF/GO) hybrid nanocomposite coated on the stainless steel mesh using sol-gel method was prepared by Abdar and co-workers [108]. The hybrid nanocomposite was used as semi-automatic SPE sorbent for the extraction of polycyclic aromatic hydrocarbons. The extracted PAHs was analyzed by gas chromatography-flame ionization detection (GC-FID). Under the optimal experimental conditions, the detection limit was 0.3-1.8 pg/mL, the relative recovery was between 95.1%-99.5%, and the relative standard deviation (RSD%) was between 4.9%-6.9%. This method combined the advantages of GO, Cu-MOF and sol-gel method, and so this hybrid nanomaterial had a variety of synergistic effects.
Moreover, an effective adsorbent of polypyrrole doped GO/COF-300 (GO/COF-300/PPy) nanocomposite was prepared for the first time by Feng and co-workers in 2022[109]. This hybrid material was used for the adsorption of indomethacin (IDM) and diclofenac (DCF) in aqueous solution. The concentration of IDM and DCF was determined by a UV-vis spectrometer at 264 nm and 274 nm. The results showed that the adsorption efficiency of GO/COF-300/PPy composite for IDM and DCF was much higher than that of single GO and COF, and the adsorption capacity of DIM and DCF were 115 mg/g and 138 mg/g, respectively. And after 8 times of adsorption-desorption process, the adsorption efficiency of GO/COF-300/PPy composite materials for IDM and DCF decreased by 27% and 29%, indicating that GO/COF-300/PPy has good reusability and stability.”
- No information available on the desorption/regeneration process of the materials used. I recommend that this be completed.
√ Many thanks for your suggestion. According to your suggestion, the desorption/regeneration process of the materials have been added to the manuscript as below, thank you very much!
“2.4 Regeneration and reproducibility of nanomaterials
When nanomaterials are used as adsorbents, the regeneration and reproducibility of the adsorbents are very important evaluation indicators.
In order to achieve good regeneration, the selection of eluent is very important, which is also the key to the optimization of solid phase extraction process. As for the reproducibility of the nanomaterial, it is necessary to carry out parallel experiments, and then calculate the relative deviation of these parallel experiments. For example, Zhao and co-workers[115] reported a novel three-dimensional interconnected magnetic chemically modified graphene oxide (3D-Mag-CMGO) for the enrichment of disperse dyes in water samples. The 3D-Mag-CMGO-adsorbed could be conveniently regenerated by using 5.0% ammonia solution/methanol (v/v) and water as eluent, allowing for them to be used repeatedly for Mag-dSPE method. After adsorption-desorption experiments, the results showed that 3D-Mag-CMGO could be reused at least ten times without much sacrifice of the extraction efficiency. In addition, six parallel experiments were carried out, and the relative deviation of the recovery rate of the target substance were between 4.6%-8.0%. In 2018[116], they tested the regeneration and reproducibility of quaternary ammonium modified magnetic carboxyl-carbon nanotubes (QA-Mag-CCNTs) and found that ammonia solution/water (5.0% v/v) was the best eluent. After desorption, the recovery rate of the target substance could reach 89.2%. And after 20 times of adsorption-desorption experiments, the adsorption efficiency of the nanomaterial to the targets did not decrease significantly. And after 8 parallel experiments, the relative standard deviation was between 3.6%-7.0%. And in the same year, Zhang and co-workers tested the regeneration and reproducibility of the three-dimensional ionic liquid-ferrite functionalized graphene oxide nanocomposite (3D-IL-Fe3O4-GO)[117]. For optimal regeneration, they conducted a lot of experiments. Results showed that the adsorbed 3D-IL-Fe3O4-GO could be conveniently regenerated by washing with toluene and water sequentially. And 3D-IL-Fe3O4-GO could be reused at least 10 times with nearly invariable quantitative efficiency. For reproducible detection, five batches of 3D-IL-Fe3O4-GO were prepared and applied in five parallel sets of experiments, the relative deviation is between 4.5% and 8.6%.”
Yours sincerely
Prof. Yong-Gang Zhao
Zhejiang Shuren University, Hangzhou 310015, China
E-mail address: zhyg91213@163.com